# Properties of Poly(3-hydroxybutyrate-*co*-3-hydroxyvalerate)/Polycaprolactone Polymer Mixtures Reinforced by Cellulose Nanocrystals: Experimental and Simulation Studies

**DOI:** 10.3390/polym14020340

**Published:** 2022-01-16

**Authors:** Marina I. Voronova, Darya L. Gurina, Oleg V. Surov

**Affiliations:** G.A. Krestov Institute of Solution Chemistry of the Russian Academy of Sciences, 1 Akademicheskaya St., 153045 Ivanovo, Russia; miv@isc-ras.ru (M.I.V.); gdl@isc-ras.ru (D.L.G.)

**Keywords:** cellulose nanocrystals, poly(3-hydroxybutyrate-*co*-3-hydroxyvalerate), polycaprolactone, polymer composites, morphology, physicochemical properties, structure–property relationship

## Abstract

Poly(3-hydroxybutyrate-*co*-3-hydroxyvalerate)/polycaprolactone (PHBV/PCL) polymer mixtures reinforced by cellulose nanocrystals (CNCs) have been obtained. To improve the CNC compatibility with the hydrophobic PHBV/PCL matrix, the CNC surface was modified by amphiphilic polymers, i.e., polyvinylpyrrolidone (PVP) and polyacrylamide (PAM). The polymer composites were characterized by FTIR, DSC, TG, XRD, microscopy, BET surface area, and tensile testing. The morphological, sorption, thermal, and mechanical properties of the obtained composites have been studied. It was found out that with an increase in the CNC content in the composites, the porosity of the films increased, which was reflected in an increase in their specific surface areas and water sorption. An analysis of the IR spectra confirms that hydrogen bonds can be formed between the CNC hydroxyl- and the –CO– groups of PCL and PHBV. The thermal decomposition of CNC in the PHBV/PCL/CNC composites starts at a much higher temperature than the decomposition of pure CNC. It was revealed that CNCs can either induce crystallization and the polymer crystallite growth or act as a compatibilizer of a mixture of the polymers causing their amorphization. The CNC addition significantly reduces the elongation and strength of the composites, but changes Young’s modulus insignificantly, i.e., the mechanical properties of the composites are retained under conditions of small linear deformations. A molecular-dynamics simulation of several systems, starting from simplest binary (solvent-polymer) and finishing with multi-component (CNC—polymer mixture—solvent) systems, has been made. It is concluded that the surface modification of CNCs with amphiphilic polymers makes it possible to obtain the CNC composites with hydrophobic polymer matrices.

## 1. Introduction

Cellulose is one of most available renewable natural resources with an annual production rate of about 1000 billion tons. As a cheap biopolymer, cellulose plays an important role in production of ecologically pure biocompatible and biodegradable functional materials. Rod-like particles of cellulose nanocrystals (CNCs) can be isolated from cellulose fibers under acid or enzymatic hydrolysis conditions. The dimensions of these particles ranges from 100 to 1000 nm in length and from 5 to 50 nm in diameter depending on hydrolysis conditions and raw material used [1,2].

At present, CNCs attract attention by material scientists not only due to their availability and ecological compatibility but also because of their unique combination of physical and chemical properties: biocompatibility, biodegradability, large specific surface area and high modulus of elasticity [3,4]. The application of CNCs as fillers in polymers allows materials to gain new quality, improving their mechanical, optical and sorption properties, electrical performance, and control humidity [5].

The application of biodegradable polymers and polymeric composite materials attracts increasing attention due to arising environmental protection issues. The high level of technological development of society is increasing the demand for biodegradable household materials (containers and packaging) and medical supplies (suture materials and implants). Besides, it is becoming more and more urgent to solve the environmental problems associated caused by plastic waste pollution. Nowadays, a tendency to use natural organic nanofillers is caused by their advantages compared to conventional inorganic fillers: biodegradability and low toxicity. The use of CNCs as nano-dimentional elements for the reinforcement of polymeric matrices is of interest due to unique combination of required physical and chemical properties and environmental benefits [6,7,8,9]. Now CNCs are widely used as a reinforcing element for developing biodegradable composites. In recent years, studies of the properties of CNC composites with polymers have made a significant contribution to the development of biodegradable and biocompatible materials, and functional materials with useful properties [10].

In terms of polymer-filler compatibility, the polymer matrix choice is always of crucial importance because the hydrophilic nature of CNCs limits its application range to hydrophilic or polar media. However, CNC’s properties make it possible to significantly modify its surface via surface hydroxyl groups possessing high reactivity. The methods of CNC surface modification can be for convenience divided into two groups: surface modification by physical adsorption of surfactants, charged particles, polymers or polyelectrolytes; and chemical modification including etherification, oxidation, introduction of silyl groups, polymer grafting, etc. [11]. The CNC surface modification can significantly expand the range of polymer matrices used for creating composites, in particular, dispersing nanocellulose in organic solvents, mixing with polymer solutions, or directly adding it to the polymer melt. However, CNC surface modification for higher stability of suspensions (which is crucial for uniform dispersion in polymer matrices) in some cases leads to a significant deterioration in the properties of CNC-based nanocomposites [5]. Such modifications make the interaction via hydrogen bonds between the nanoparticles much weaker, and prevents improving the mechanical properties of the composites. New approaches must be developed to creating nanocomposites that combine better mechanical properties and higher thermal stability, which will allow treating them by economically viable industrial methods, such as extrusion molding or pressure casting, which are normally used for the molding of thermoplastic polymers.

Among the common polymers used to prepare biodegradable materials are, for instance, polylactic and polyglycolic acids, copolymers of lactic and glycolic acids, polycaprolactone (PCL), and poly(3-hydroxybutyrate-*co*-3-hydroxyvalerate) (PHBV). PCL is an aliphatic semicrystalline polyester well-soluble in a number of organic solvents, and is compatible with other biodegradable polymers that promote its application in regenerative medicine. PHBV is a natural biopolymer which can be obtained by microbiological methods. Both polymers have good biocompatible, bioresorbable, and biodegradable properties; they are used as materials for the manufacture of polymer medical devices, implants for substitution surgery, in tissue engineering, and in drug delivery systems.

PCL and PHBV are capable of enzymatic hydrolytic degradation in living organisms, have excellent film-formation properties, and are soluble in the same solvents, which will make it possible to obtain materials with a specified structure and functional characteristics through composition variations. PHBV and PCL are biocompatible, can retain their functional properties but can decompose to natural metabolites in natural conditions and in living organisms. This means that it is possible, based on them, to develop materials contacting with living tissues, but polymer composites developed for every application field are expected to possess a specified complex of properties.

Preparation of PCL/CNC and PHBV/CNC composites has been described in several works. Homogeneous distribution of CNCs in a PCL hydrophobic matrix is achieved by modifying the surface of CNC particles [12] or surface modification of PCL through plasma chemical treatment [13,14], using a CNC organogel [15] or CNC solution in a suitable solvent [16] including an ionic liquid [17]. PHBV/CNC composites can be prepared by CNC surface graft modification and melt blending [18], by means of electrospinning [19], using a Pickering emulsion approach [20], using different compounding [21,22,23] or extrusion methods [24], or via a simple solution casting method [25,26,27].

Although PCL and PHBV are used in medicine as a suture and tissue engineering material, as well as a durable bioresorbable thermal implant, their uses are limited by high surface hydrophobicity that prevents cell integration with the matrix. Moreover, pure PHBV is very fragile due to its high crystallinity, and is characterized by low thermal stability. On the other hand, neat PCL is an elastic homopolymer with a low melting point. The creation of composites based on mixtures of PHBV and PCL reinforced by CNCs would make it possible to level the disadvantages of each individual polymer and significantly expand the application area of polymeric systems. Nevertheless, as far as we know, there are currently no works in the literature describing the properties of a PHBV/PCL polymer mixture reinforced with CNCs.

The purpose of the work is to obtain CNC composites based on a mixture of hydrophobic biocompatible polymers PHBV/PCL and to study their morphological, mechanical, thermal, and sorption properties. To this aim, CNC modification through physical adsorption of water-soluble amphiphilic polymers (polyvinylpyrrolidone (PVP) and polyacrylamide (PAM)) was performed. The making of PCL/PHBV/CNC composites involves preparing a stable suspension of CNCs in a solvent in which the polymers can be dissolved or which can be mixed with a solvent appropriate for the polymers. Modification of CNC surface by adsorption of amphiphilic polymers is known to improve redispersibility in organic solvents and compatibility with hydrophobic matrices [28]. We have shown in our earlier studies [29,30] that adsorption of PVP or PAM on CNC particles can significantly improve the redispersibility of CNCs in some organic solvents, including dichloromethane and chloroform. It has been shown that the adsorption of PVP on the surface of CNCs leads to steric hindrances that prevent the interactions between particles of the CNCs and contribute to the good redispersity of lyophilized modified CNCs in water and some solvents. Owing to its amphiphilic nature, PVP can act as a surface stabilizer and dispersant of CNC nanoparticles, preventing their aggregation due to the repulsive forces created by its hydrophobic carbon chains propagating into solvents and interacting with each other [31]. It was shown that PAM globule adsorption onto CNC particles caused CNC surface hydrophobization and a decrease in their surface charge, while maintaining high colloidal stability of the CNC suspensions. Thus, the PAM globule conformation provided good re-dispersibility of the freeze-dried modified CNCs by preventing aggregation of the CNC particles [32]. We hypothesize that this approach will allow us to obtain CNC composites with hydrophobic polymers successfully. Studying the properties of the composites will widen the application range of biodegradable composites based on CNCs and the hydrophobic biocompatible polymers PCL and PHBV.

Moreover, a computer simulation of the systems hydrophilic polymer/CNC (exemplified by PVP), hydrophobic polymer/CNC (exemplified by a PHBV/PCL mixture) were conducted in the work. The simulation results made it possible to identify and better understand the mechanisms of CNC interaction with hydrophilic and hydrophobic polymers constituting a composite. The obtained results allowed more knowledge about the compatibility of physically modified CNCs with hydrophobic polymer matrixes to be achieved.

## 2. Materials and Methods

### 2.1. Materials

Microcrystalline cellulose (MCC) (powder ~20 mm), dichloromethane (DCM) (CH_2_Cl_2_, chemically pure grade), PAM (MW 40,000), PVP (MW 40,000), PCL (MW 80,000), and PHBV (natural origin, PHV content 12 mol%) were purchased from Sigma-Aldrich (Saint Louis, MO, USA). Sulfuric acid (H_2_SO_4_, chemically pure grade) was purchased from Chimmed (Moscow, Russia). 

### 2.2. Methods

#### 2.2.1. Preparation of CNCs

The aqueous suspensions of CNCs were prepared by hydrolysis of MCC in sulfuric acid as described earlier [33]. The hydrolysis was carried out in a 62% aqueous solution of sulfuric acid (1 g of MCC in 40 mL of the solution) at 50 °C during 2 h under vigorous stirring. The resulting suspension was rinsed with distilled water to remove the acid by repeated centrifugation until a constant pH (~2.4) of the supernatant was achieved. Next, the CNC suspension was purified (ion-exchange resin TOKEM MB-50(R), Kemerovo, Russia), sonicated (Sonorex DT100, Bandelin, Berlin, Germany) for 15–30 min, and used for PVP- or PAM-modification. 

#### 2.2.2. Preparation of PVP- and PAM-Modified CNCs

The modified CNCs were prepared by mixing 0.5 g of PAM (or PVP), 10 mL of distilled water and the required amount of an aqueous CNC suspension to achieve the PAM (or PVP) content of 10 wt.%. Then the modified CNC samples were lyophilized. Before the lyophobization, the samples were stored frozen at −40 °C for 2 days. Then they were placed in a freeze drying chamber. The drying took place for 48 h at a pressure of 6 Pa, and a temperature of −54 °C. The lyophilized modified CNC samples were dispersed in DCM for 2 h with vigorous stirring followed by 30 min of ultrasonic treatment, and used for production of PHBV/PCL/CNC composite films. 

#### 2.2.3. Preparation of PHBV/PCL/CNC Composite Films

Two series of composites with different ratios of polymers PHBV and PCL (1:1 and 1:2) were obtained. To obtain composites with 1:1 polymer ratio, 0.5 g of PHBV and 0.5 g of PCL were stirred in 10 mL of a CNC suspension in DCM for 2 h at room temperature to dissolve the components. To obtain composites with 1:2 polymer ratio, 0.25 g of PHBV and 0.5 g of PCL were used, respectively. The prepared mixtures were cast into glass Petri dishes and dried at room temperature. The samples obtained were designated as PHBV/PCL(1:1), PHBV/PCL(1:1)/CNC(PVP)–5, PHBV/PCL(1:2)/CNC(PAM)–10, etc., where 5, 10 is the content of CNCs in the composite, wt.%; 1:1 or 1:2 is the ratio of PHBV and PCL in their mixture; PVP or PAM is an amphiphilic polymer for the surface modification of the CNC particles.

### 2.3. Characterization

To investigate the morphology of the samples, a VEGA3 TESCAN scanning electron microscope (SEM) (Brno, Czech Republic) and a Soptop CX40P polarization optical microscope (POM) (Sunny Instruments, Ningbo, China) were employed. 

To determine the specific surface area of the samples, a NOVAtouch NT LX Quantachrome (Boynton Beach, FL, USA) automatic analyzer was applied. Before measurements, the samples were kept for 3 h in a vacuum at 50 °C. The low-temperature (–196 °C) isotherms of nitrogen adsorption and desorption were measured at relative pressure of 0.01–0.95. The specific surface areas of the samples under study were calculated by the Brunauer–Emmett–Teller (BET) method. 

The isotherms of water adsorption and desorption were registered under controlled humidity at 25 °C. The values of water uptake were determined gravimetrically.

The FTIR spectra were recorded in the range of 4000–400 cm^−1^ using a VERTEX 80v spectrophotometer (Bruker, Ettlingen, Germany). The samples with 1 mg of the compound to be analyzed, together with 100 mg of potassium bromide, were pressed into tablets.

The DSC measurements were performed on a DSC 204 F1 (Netzsch, Selb, Germany) in an atmosphere of ultrapure grade dry argon at a flow rate of 15 mL min^−1^ and a heating rate of 10 K min^−1^ using standard aluminum crucibles. The degree of crystallinity (*χ*_c_) of the samples was calculated by the following equation:*χ*_c_ = Δ*H*_m_/*w*Δ*H*_m_^0^,(1)
where Δ*H*_m_ is the heat of fusion measured from DSC thermograms, *w* is the weight fraction of a semicrystalline polymer in a polymer binary blend or in a CNC-based composite, Δ*H*_m_^0^ is the heat of fusion of the 100% crystalline polymer. Δ*H*_m_^0^ values reported in previous works are 157 and 146.6 J g^−^^1^ for PCL and PHBV, respectively [34,35].

The TG analysis was performed using a TG 209 F1 Iris thermomicrobalance (Netzsch, Selb, Germany) with platinum crucibles in a dry argon atmosphere at a flow rate of 30 mL min^−1^, and a heating rate of 10 K min^−1^.

The X-ray diffraction analysis was performed using a Bruker D8 Advance powder diffractometer (Bruker, Rheistetten, Germany) according to the Bragg–Brentano scheme with Cu-*K*α radiation (λ = 0.1542 nm). The crystallinity index was determined by the Segal method. The crystallinity of PHBV and PCL was calculated in Diffrac.Suite program package.

The tensile properties of the sample films were evaluated using an I 1158 M-2.5-01-1 tensile-testing machine (Ivanovo, Russia) at a loading rate of 1 mm per min^−1^ in the tension mode. 

### 2.4. Computational Details 

The classical molecular dynamics (MD) simulation was carried out using the GPU version of GROMACS 5.0.7 [36]. For the PVP macromolecule (Figure 1a), we applied the united atoms model based on GROMOSG53a6 parameters, which had been earlier employed to study the PVP-cellulose-water system [37]. For the PCL macromolecule (Figure 1b), we used the united atoms model provided in [38]. The PCL and PVP macromolecules have a comparable chain length, although their degree of polymerization is different. We used Avogadro to construct the initial structure of the PCL and PVP macromolecules containing 10 and 60 monomer units (with the molecular weights of 1162.6 and 6668.6 g/mol, respectively) [39]. Due to the fact that commercially available PHBV grades only have up to 20 mol% hydroxyvalerate content [40], we have constructed the polymer chain containing from 18 hydroxybutyrate units and 2 hydroxyvalerate units (with the molecular weight of 1838 g/mol) by means of Avogadro as well (Figure 1c). In order to generate united atom topology of PHBV macromolecule based on Gromos 54A7 parameters we used Automated Topology Builder [41]. The initial structure of a DCM molecule and potential parameters were taken from the Automatic Topology Builder [41] as well. For water molecules, we applied the SPC/E (Extended Simple Point Charge) model [42]. For cellulose, GROMOS54a7 force field parameters were used [41]. The initial structure of a CNC particle (as a model of Iβ cellulose) was built with the Cellulose Builder toolkit [43] based on the experimental crystallographic data [44], and consisted of 9 glucan chains with the degree of polymerization of 10. The *NpT* simulations were performed at 0.1 MPa (Parrinello-Rahman barostat) and 298 K (Nose-Hoover thermostat [45,46]. For long-range interactions with a cutoff distance of 1.5 nm (both the van der Waals and the electrostatic interactions), the particle-mesh Ewald (PME) method was used [47,48]. The data for analysis were collected every 0.1 ps with the time step of 1 fs. 

## 3. Results and Discussion

### 3.1. Morphological and Adsorption Properties of PHBV/PCL/CNC Composites

Structural features of polymeric materials largely determine their consumer properties and prospects for use in a particular area. The use of pure PCL and PHBV as materials is limited by their high hydrophobicity, low melting points, and low mechanical strength. An effective way to regulate the supramolecular, morphological and porous structure of materials based on PCL and PHBV may be the formation of composite films from their mixed solutions, as well as the use of fillers and reinforcing additives. Establishing the relationship between the composition, morphology and properties of films is an important task and the basis for obtaining composites with the required consumer properties. The properties of PHBV/PCL mixtures have been studied in several works [49,50,51]. In our work, we investigated the features of PHBV/PCL/CNC composite films. The use of PVP- or PAM-modified CNC allows PHBV/PCL/CNC composite films to be prepared by casting from a solution in DCM. Figure 2, Figure 3 and Figure 4, Appendix A show SEM and POM images of the surfaces and cleavages of the PHBV/PCL/CNC composite films with various contents of modified CNCs (5, 10 and 15 wt.%). For comparison, the figures also show images of the PHBV/PCL films (Appendix A).

It can be seen that the surfaces of the PHBV/PCL/CNC composite films are inhomogeneous, with a pronounced porosity. As shown by cross section images, a porous structure is characteristic of the entire volume of the films. With an increase in the CNC content in the composites, the porosity of the films increases, which is reflected in an increase in their specific surface areas and water sorption (Table 1). In polarized light, it can be seen that pure PCL crystallizes in the form of large spherulites. The formation of a mixture with PHBV, as well as the addition of CNCs, leads to a considerable decrease in the PCL spherulite size and then to their complete disappearance (Figure 4, Appendix A).

Figure 5 and Figure 6 show isotherms of low-temperature nitrogen adsorption and desorption on PHBV/PCL and PHBV/PCL/CNC films. The isotherms are almost straight lines without hysteresis between the adsorption and desorption branches, the shape of which is attributed to adsorption on a macroporous adsorbent.

An increase in CNC content in the composites leads to a significant increase in water sorption (Figure 7 and Figure 8). As stated by their shape, the water adsorption isotherms may be assigned to type II isotherms according to Brunauer’s classification [52]. The isotherms under study are characterized by a significant increase in water vapor adsorption in the region of increased relative pressures and practically no hysteresis. This behavior is inherent in polymolecular adsorption on macroporous hydrophobic adsorbents [53].

### 3.2. FTIR Analysis

For neat PCL, main characteristic bands are observed in the FTIR spectra: 2945 and 2864 cm^−1^ (asymmetric and symmetric vibrational vibrations of CH_2_), 1729 cm^−1^ (C=O vibrations), 1240 and 1166 cm^−1^ (asymmetric and symmetric C–O–C vibrations). For neat PHBV, the absorption bands at 1280, 1456, 1723, 2930, and 2979 cm^−1^ refer to stretching vibrations –COC–, bending vibrations –CH_2_–, stretching vibrations –CO– in ester, and symmetric and asymmetric stretching vibrations –CH_3_, respectively [54] (Appendix A). In the FTIR spectra of PHBV/PCL mixtures, an overlap of spectral bands characteristic for pure PHBV and PCL is observed [49,50,51]. FTIR spectra of CNC are characterized by intense absorption bands in the range of wavenumbers 3300–3500 cm^−1^ and 2800–3000 cm^−1^, which correspond to stretching vibrations of OH–groups and CH–bonds of cellulose. The absorption region 950–1200 cm^−1^ is assigned to valence vibrations of C–O, C–C of a pyranose ring structure [29]. In the PHBV/PCL/CNC composites, the spectral bands mostly overlap, and that fact complicates their interpretation. In addition, cellulose characteristic bands do not appear due to CNC low content in the composite. However, CNC incorporation into PHBV/PCL polymer matrix leads to broadening of the stretching vibration bands at 3300 cm^−1^ and 1730 cm^−1^, responsible respectively for –OH and C=O stretching vibrations, and may be associated with the hydrogen bond formation (Figure 9a and Appendix A). In addition, it should be noted that with an increase in the CNC content, the 1728 cm^−1^ band characterizing vibrations of the –CO– group involved in hydrogen bonds, shifts towards lower wavenumbers, while the 1746 cm^−1^ band (characterizes the vibrations of free unbound groups –CO–) shifts towards higher ones. This may serve as an evidence for the hydrogen bond formation between the CNC hydroxyls and the –CO– groups of PCL and PHBV (Figure 9b and Appendix A).

### 3.3. Thermal Properties 

Thermal characteristics of the PHBV/PCL/CNC composites were obtained based on the analysis of differential scanning calorimetry (DSC) and thermogravimetry (TG) data. DSC curves were obtained at heating and cooling (Figure 10, Figure 11 and Appendix A). The values of the melting temperature *T*_m_, the crystallization temperature *T*_cryst_, the heat of fusion (Δ*H*_m_) and the degree of crystallinity (*χ*_c_) of the polymers in the composites with CNCs are given in Table 2. The pure PCL melts at 55.6 °C and crystallizes at 24.1 °C. In the composites under study, the PCL melting point practically does not change, and the crystallization temperature varies within 30 °C. DSC curves for the pure PHBV show two melting endothermic peaks: low-temperature (*T*_m1_ = 145.7 °C) and high-temperature (*T*_m2_ = 156.5 °C). The appearance of the high-temperature peak is attributed to the melting of more perfect polymer crystals formed as a result of primary crystallization, while the low-temperature peak is attributed to the melting of less structured crystals formed by secondary crystallization [18]. The PHBV/PCL/CNC composites are also characterized by double melting peaks, but their position and intensity differ from the pure PHBV. The intensity of the low-temperature peak slightly decreases, the temperature *T*_m1_ tends to decrease. The intensity of the high-temperature peak remains almost unchanged, as the melting point *T*_m2_ does, while the crystallization temperature tends to decrease. 

Changes in the degree of crystallinity of the polymers are complicated: in the PHBV/PCL(1:1) mixture, an increase in the CNC content in the composites increases the degree of crystallinity of the polymers, while in the PHBV/PCL(1:2) mixture the opposite trend is observed. Thus, under different conditions, CNCs can act both as an initiator of crystallization and growth of polymer crystallites and as a compatibilizer of polymer mixtures, causing their amorphization and improving miscibility [55].

The thermogravimetric (TG) curves and corresponding derivative thermogravimetry (DTG) curves of the PHBV/PCL mixtures and PHBV/PLC/CNC composites are shown in Figure 12 and Appendix A, and the appropriate characteristic data are listed in Table 3. 

The TG curves of the PHBV/PCL mixtures contain two regions of weight loss corresponding to the decomposition of the individual polymers (Appendix A). The TG analysis data show a slight increase in the decomposition temperature of PHBV in a mixture with PCL as compared to pure PHBV, while the decomposition temperature of PCL remains practically unchanged. It should be noted that the thermal decomposition of CNCs in the composites occurs at a significantly higher temperature than the decomposition of pure CNCs, for which *T*_on_ = 200 °C and *T*_max_ = 240 °C [56] (Appendix A). Only in the case of the PVP-modified CNCs on the TG curves can one observe the appearance of a low-temperature shoulder, responsible for the CNC decomposition (Figure 12 and Appendix A). 

### 3.4. X-ray Diffraction Analysis

Appendix A shows X-ray diffraction patterns of neat PHBV and PCL, and their mixtures in 1:1 and 1:2 ratios. The diffractogram of pure PCL is characterized by two reflections at Bragg angles of about 21.0° and 23.1°, which correspond to the crystallographic planes (110) and (200) of the orthorhombic PCL lattice [57], respectively. The characteristic diffraction peaks of PHBV are located at the Bragg angles of 13.3, 16.8, 21.4, 25.6, and 29.9°, corresponding to the crystallographic planes (020), (110), (111), (121), and (002), respectively [58]. The X-ray diffraction patterns of the polymer mixtures are superpositions of the diffraction patterns of the pure polymers. In the diffraction patterns of the composites, the diffraction peaks of CNC (2θ = 22.9; 16.6; 14.8°), corresponding to the crystallographic planes (200), (110), and (1–10) of cellulose I_ß_, respectively [59], are practically invisible due to the low CNC content and the complexity of the multicomponent system (Figure 13 and Appendix A). 

The total crystallinity of PHBV and PCL polymers in the composites with CNCs is shown in Table 4. The data of X-ray diffraction analysis mainly confirm the DSC data about complicated dependence of crystallinity of the polymers on the CNC content in the composites.

### 3.5. Tensile Properties 

Neat PCL is an elastic polymer characterized by a very high relative deformation at break (over 700%). On the contrary, neat PHBV is a very brittle polymer, the relative elongation at break of which does not exceed 1.5% (Appendix A). The mixtures of the polymers are characterized by a significant decrease in the elongation at break with the retention of strength characteristics (Table 5). The CNC addition significantly reduces the elongation and strength, i.e., the composite material becomes less elastic and more brittle (Appendix A). At the same time, with an increase in the CNC content, Young’s modulus changes insignificantly, which indicates that the mechanical properties of the composites are retained under conditions of small linear deformations (Table 5). 

### 3.6. Molecular Dynamics Simulation 

#### 3.6.1. Intermolecular Interactions in a Multicomponent System Containing CNC, PVP, PHBV, and a Solvent (Water, DCM)

The simulation details of the systems consisting of CNC, PVP, PHBV, and a solvent are presented in Table 6.

Figure 14 shows the initial configuration of the PHBV macromolecule and its conformation after 1 ns simulation in vacuum.

In DCM, the average values of the end-to-end distance and radius of gyration of the PHBV are, respectively, <R_e-t-e_> = 3.49 ± 1.05 nm, <R_g_> = 1.35 ± 0.25 nm, and in vacuum <R_e-t-e_> = 1.01 ± 0.43 nm, <R_g_> = 0.69 ± 0.04 nm (Figure 15). In water, the average values are <R_e-t-e_> = 1.24 ± 0.40 nm, <R_g_> = 0.68 ± 0.03 nm.

Despite its hydrophobic nature, the PHBV macromolecule is capable of forming hydrogen bonds (HBs) with water molecules. The average amount of HB per one monomer unit of the polymer is 0.50.

A cellulose nanoparticle with adsorbed PHBV macromolecule was placed in a cubic box filled with DCM molecules. Since DCM is a good solvent for PHBV, it is supposed that the polymer can be desorbed from the CNC surface. To verify this suggestion, the number of close contacts of any CNC atoms with PHBV within a distance of 0.5 nm as well as the number of HBs formed by the polymer and CNC was calculated as a function of time (Figure 16). It is seen that, during the first 4–5 ns, the PHBV is completely desorbed from the CNC surface and goes into the solution bulk.

Figure 17 and Figure 18 show that placing the PHBV/PVP/CNC composite (consisting of one cellulose nanoparticle and 13 PVP macromolecules and 8 PHBV macromolecules adsorbed on it) into DCM (Table 6, System 8) triggered the desorption process similar to the process for PCL/PVP/CNC [33]. Figure 17 shows a rapid increase in the radius of gyration and end-to-end distance of PVP and PHBV in the first nanosecond of the simulation process, with further fluctuation near constant values: <R_e-t-e_> = 4.54 ± 0.15, <R_g_> = 1.71 ± 0.03 and <R_e-t-e_> = 3.43 ± 0.24, <R_g_> = 1.30 ± 0.05 nm for PVP and PHBV, respectively.

The PVP and PHBV macromolecules, which were at a greater distance from the CNC and did not have direct contacts with it, were desorbed first and passed into the solution. The PVP molecules that were in direct contact with the CNC were not completely desorbed, although the number of PVP–CNC contacts decreased significantly by the end of the simulation (Figure 18). The number of PVP–PHBV contacts during the composite dissolution in DCM also decreased sharply and reached an average value of 114 (Figure 18e). The number of contacts between the polymers of the same type also rapidly decreased in the first 5 ns, and then N_C_ fluctuated near a constant value. It should be noted that the number of PVP–PVP and PHBV–PHBV contacts (Figure 18c,d) includes not only the contacts between different macromolecules, but also those between the atoms of one (the same) macromolecule; therefore, these values of N_C_ are quite large compared to those for pairs polymer–CNC or PVP–PHBV (Figure 18a,b,e).

Figure 19, which shows the time dependence of the polymer–cellulose HB number, also indicates that the PVP molecules were not completely desorbed from the CNC surface, and by the end of the simulation, the polymer formed about 17.8 HBs with the hydroxyl groups of the cellulose nanoparticles. At the same time, the number of HBs formed by the PHBV with the cellulose is much lower than that of the PVP–CNC HBs, which once again confirms the fact of the low affinity of the hydrophobic polymer (PHBV) and hydrophilic cellulose nanoparticles (CNCs).

Figure 20 shows snapshots of the simulation cells at the beginning and end of the simulation.

#### 3.6.2. Intermolecular Interactions in Binary PHBV/PCL System in DCM and Vacuum

Modeling of three systems was carried out, with low and high polymer concentration in DCM (Systems 1 and 2, respectively), as well as modeling of a composite consisting of PCL and PHBV (in vacuum) (System 3). The simulation details of the systems consisting of PCL, PHBV, and DCM are presented in Table 7. Figure 21 shows the snapshots of the Systems 1–3 at the end of the simulation.

Based on the simulation results, average values of the radius of gyration and end-to-end distance of the polymer macromolecules were calculated (Table 8), as well as the number of close contacts of the polymer atoms within 0.5 nm (Table 9) (averaging was carried out over the last 5 ns of the simulation).

DCM is a good solvent for PCL and PHBV, as evidenced by the R_g_ and R_e-t-e_ values. In a dilute solution, the R_g_ and R_e-t-e_ values are slightly higher than in a concentrated solution of the polymer mixture (Table 8). At low concentrations, PCL and PHBV macromolecules in DCM practically do not interact with each other: the number of close contacts over the last 5 ns of the simulation is 0 (Table 9, System 1).

After the removal of DCM from the solution with a high content of the polymers, the number of PCL–PHBV close contacts increased more than 38 times (Table 9, System 3). Compared with the R_e-t-e_ and R_g_ values of individual PCL and PHBV macromolecules in vacuum (<R_e-t-e_> = 0.82 ± 0.23 nm, <R_g_> = 0.71 ± 0.05 nm, and <R_e-t-e_> = 1.01 ± 0.43 nm, <R_g_> = 0.69 ± 0.04 nm, for PCL and PHBV, respectively), the R_e-t-e_ and R_g_ values for PCL and PHBV (System 3) are significantly higher. That is, in the polymer mixture, the PCL and PHBV macromolecules are not in the form of a dense globule, but in the conformation of expanded chains (as in a concentrated solution). It should be noted that not only atoms of different polymer contribute to the value of N_C_ (Table 9), but also atoms belonging to the same polymer chain are taken into account. Therefore, the number of PCL–PCL and PHBV–PHBV contacts in System 1 is not equal to 0. The calculated density of the final polymer mixture in vacuum (System 3) was equal 1081 ± 7 kg/m^3^. The experimental density of the pure polymers is 1245 kg/m^3^ [60] and 1145 kg/m^3^ for PCL and PHBV, respectively.

The radial distribution functions (RDFs) between the centers of mass of PCL and PHBV macromolecules in DCM and vacuum were calculated (Figure 22). In DCM, the first RDF maxima are at 2.0, 1.5, and 0.5 nm for the interactions of PCL–PCL, PHBV–PHBV, and PCL–PHBV, respectively. In vacuum, the first maxima at the corresponding RDFs are observed at shorter distances of 0.59, 0.76, and 0.28 nm, respectively. So, both in System 2 and in System 3, the PCL–PHBV interactions are more preferable than the PCL–PCL or PHBV–PHBV interactions.

#### 3.6.3. Intermolecular Interactions in a Multicomponent System Containing CNC, PVP, PHBV, PCL, and a Solvent (Water, DCM)

Cellulose nanoparticle with adsorbed PVP was placed in the center of a cubic box with periodic boundary conditions. After that, 10 molecules of PCL and 10 molecules of PHVB were added randomly to the box with the CNC/PVP sample. This system was then put into an *NpT* ensemble with a pressure of 1 MPa and a temperature of 500 K for 1 ns with a simulation time step of 1 fs. This step allows the composite structure to be slowly compressed to obtain the initial amorphous matrix. Then, the temperature was set to 298 K and a pressure of 0.1 MPa for 10 ns. The density of the final matrix was 1.17 g/cm^3^. The final configuration of the composite obtained at the previous step was placed in the center of a cubic box with periodic boundary conditions containing 11,000 DCM molecules. The simulation was performed in *NpT* ensemble at 298 K and 0.1 MPa for 10 ns.

Figure 23 shows an increase in the number of close contacts between atoms of the composite components in the initial simulation period at 500 K and 1 MPa and in the equilibrium period at 298 K and 0.1 Pa. In the first 100 ps, the initial decrease of R_g_ and R_e-t-e_ values for PCL and PHVB with achieving their minima is observed (inset in Figure 24). This behavior is due to the folding of PCL and PHBV macromolecules into globules in vacuum. After 100 ps, the R_g_ and R_e-t-e_ values begin to rise and reach a plateau. This growth is associated with the transition of PCL and PHBV macromolecules to the conformation of expanded chains as a result of interaction with PVP on the CNC surface. It should be noted that, in comparison with PCL, PHBV has a larger number of close contacts with PVP, i.e., interacts better with it.

In Figure 25 the structure of final configuration of the composite PHVB/PCL/PVP/CNC is presented (the simulated cell and its periodic images).

As DCM is a good solvent for the polymers, in a DCM solution they are desorbed from the surface of the composite. In Table 10 the values of the end-to-end distance and radius of gyration of the polymers in vacuum, water, and DCM are compared. In vacuum, the R_e-t-e_ and R_g_ values for the polymer macromolecules in the composite are higher compared to the values for individual macromolecules. At the same time, in DCM, the R_e-t-e_ and R_g_ values for the polymer macromolecules in the composite are only slightly lower than the values for the individual macromolecules in the solvent. This suggests that not all macromolecules are completely desorbed from the composite surface and pass into the solution bulk by the end of the simulation, and a small part of them form HBs with the cellulose nanoparticle. Table 11 shows that average number of HBs significantly decreases for pairs CNC–PCL and CNC–PHVB. However, for the CNC–PVP pair, the <nHB> value remains almost the same. In vacuum, the R_g_ values for PCL and PHBV as individual macromolecules are practically equal; however, in the composite, the R_g_ value for PHBV is greater than that for PCR. This may indicate that, in comparison with PCL, PHBV interacts more intensively with PVP on the CNC surface, and its macromolecules in the composite are in more expanded conformation than PCL macromolecules.

Figure 26 shows the initial and final configurations of the PHBV/PCL/PVP/CNC composite in DCM. It can be seen that by the end of the simulation, the PCL, PHBV, and most of the PVP macromolecules passed into the solution bulk.

The time dependence of the number of close contacts between any pair of CNC–polymer, polymer–polymer, CNC–DCM, polymer–DCM atoms confirms the previously observed behavior of the composite in the solvent. The solvent molecules tend to penetrate the interior of the composite. The high affinity of PCL, PHBV, and PVP with the solvent promotes the dissolution of the composite. This is evidenced by an increase in the number of close contacts between the components of the system and the solvent (Figure 27b) and a decrease in the number of close contacts in polymer–polymer and CNC–polymer pairs (Figure 27a).

Figure 28 shows the radial distribution functions (RDFs) between the centers of mass of the composite components in vacuum. The maxima on the RDF curves for the PCL–PCL and PVP–PCL pairs are in the range of 0 to 0.5 nm. Comparing the RDFs for the pairs PHBV–PHBV and PCL–PCL, we can say that PCL macromolecules in the composite have a denser packing. This suggests that, compared with PHBV, PCL has a closer contact with PVP. In DCM, all the peaks on the RDFs shift towards larger distances, indicating the desorption process and the movement of the polymer macromolecules into the solution bulk (Figure 29).

## 4. Conclusions

CNC composites with PCL and PHBV with different PHBV/PCL ratios (1:1 and 1:2) have been obtained. In order to improve the CNC compatibility with the hydrophobic PCL/PHBV matrix, the CNC particle surface was modified by amphiphilic PVP and PAM polymers. The morphological, sorption, thermal, and mechanical properties of the obtained composites have been studied. An analysis of the IR spectra confirms that a hydrogen bond can be formed between the CNC hydroxyl- and the -CO- groups of PCL and PHBV. The thermal characteristics of the PHBV/PCL/CNC composites have been determined based on the DSC and TG data analysis. It is noted that the thermal decomposition of CNC in the PHBV/PCL/CNC composites starts at a much higher temperature than the decomposition of pure CNC. Based on the DSC and XRD data analysis, it is concluded that, depending on the conditions, CNCs can either induce crystallization and polymer crystallite growth or act as a compatibilizer of a mixture of polymers causing their amorphization and improving miscibility. The mixture of PHBV/PCL polymers has a much lower relative elongation at break at the same strength characteristics. At the same time, an increase in the CNC content changes Young’s modulus insignificantly, which means that the composite mechanical properties remain unchanged in conditions of small linear deformations. A molecular-dynamics simulation of several systems, starting from simplest binary (solvent–polymer) and finishing with multi-component (CNC—polymer mixture—solvent) ones, has been made in order to study the mechanisms of CNC interaction with the polymers in a composite and identify the specific features of the interactions between the system components. The obtained results make it possible to improve the CNC compatibility with hydrophobic polymer matrices and identify and study in more detail the mechanisms of CNC interaction with hydrophilic and hydrophobic polymers in a composite.

## Figures and Tables

**Figure 1 polymers-14-00340-f001:**
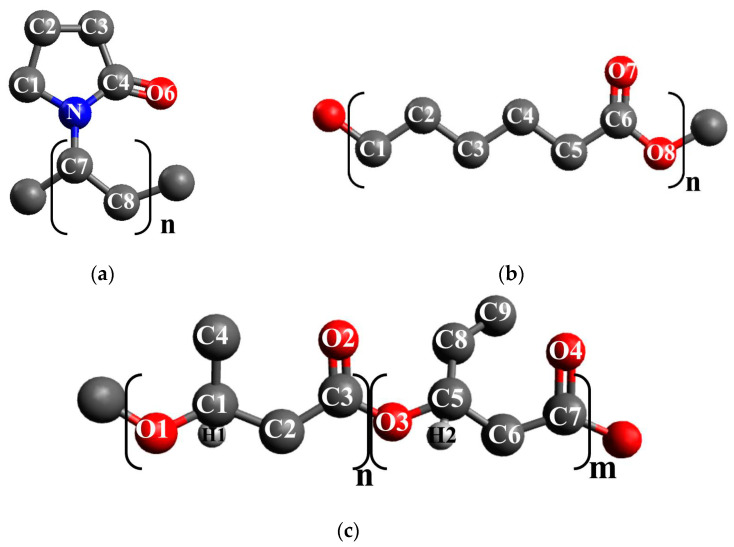
(**a**) Monomer units of PVP, (**b**) PCL, and (**c**) PHBV with atom designations.

**Figure 2 polymers-14-00340-f002:**
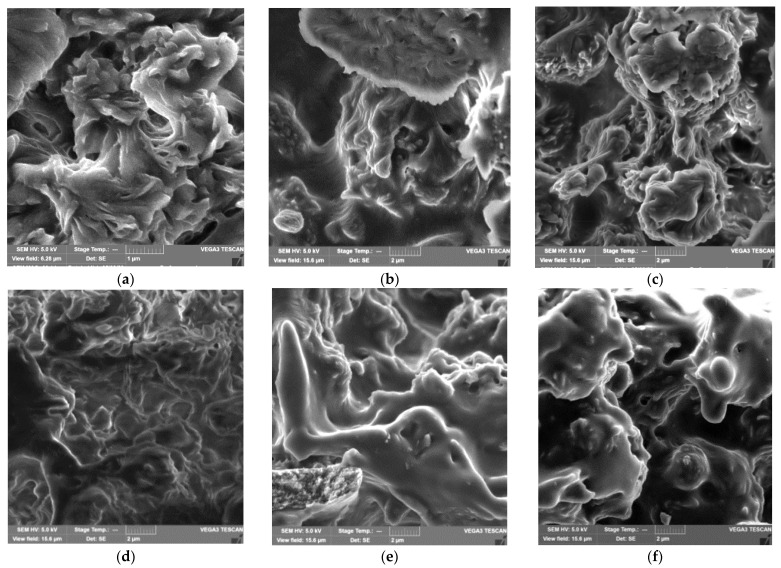
SEM images of the surface (**a**–**c**) and cleavage (**d**–**f**) of the PHBV/PCL/CNC composite films: PHBV/PCL(1:1)/CNC(PVP)–5 (**a**,**d**); PHBV/PCL(1:1)/CNC(PVP)–10 (**b**,**e**); PHBV/PCL(1:1)/CNC(PVP)-15 (**c**,**f**). The scale is 1 μm (**a**); 2 μm (**b**–**f**).

**Figure 3 polymers-14-00340-f003:**
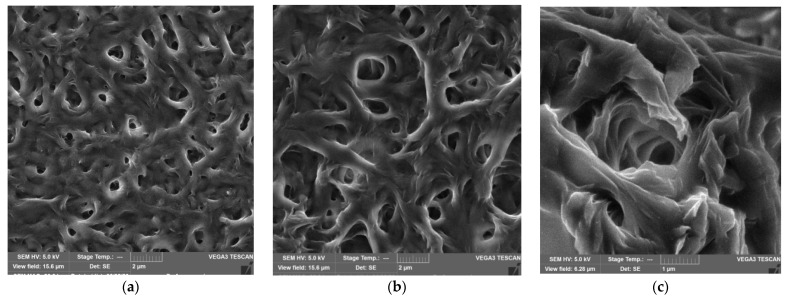
SEM images of the surface (**a**–**c**) and cleavage (**d**–**f**) of the PHBV/PCL/CNC composite films: PHBV/PCL(1:1)/CNC(PAM)–5 (**a**,**d**); PHBV/PCL(1:1)/CNC(PAM)−10 (**b**,**e**); PHBV/PCL(1:1)/CNC(PAM)–15 (**c**,**f**). The scale is 2 μm (**a**,**b**,**d**,**e**); 1 μm (**c**,**f**).

**Figure 4 polymers-14-00340-f004:**
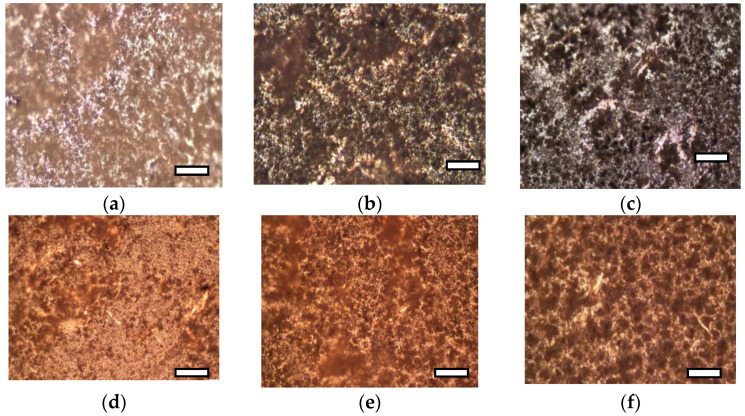
POM images of the surface of the PHBV/PCL/CNC composite films: PHBV/PCL(1:1)/CNC(PVP)–5 (**a**); PHBV/PCL(1:1)/CNC(PVP)–10 (**b**); PHBV/PCL(1:1)/CNC(PVP)–15 (**c**); PHBV/PCL(1:1)/CNC(PAM)–5 (**d**); PHBV/PCL(1:1)/CNC(PAM)–10 (**e**); PHBV/PCL(1:1)/CNC(PAM)–15 (**f**). The scale is 100 μm.

**Figure 5 polymers-14-00340-f005:**
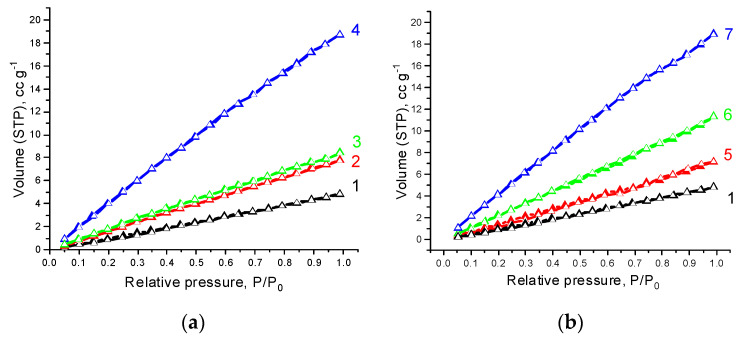
Low-temperature nitrogen adsorption (filled symbols) and desorption (open symbols) isotherms on the samples: 1—PHBV/PCL(1:1), 2—PHBV/PCL(1:1)/CNC(PVP)–5, 3—PHBV/PCL(1:1)/CNC(PVP)–10, 4—PHBV/PCL(1:1)/CNC(PVP)–15 (**a**); 5—PHBV/PCL(1:1)/CNC(PAM)–5, 6—PHBV/PCL(1:1)/CNC(PAM)–10, 7—PHBV/PCL(1:1)/CNC(PAM)–15 (**b**).

**Figure 6 polymers-14-00340-f006:**
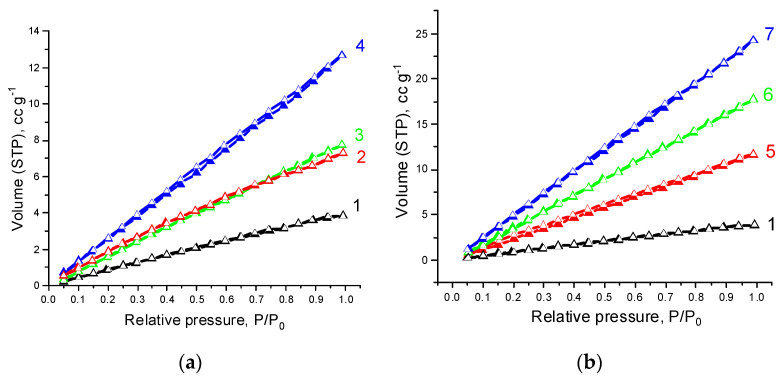
Low-temperature nitrogen adsorption (filled symbols) and desorption (open symbols) isotherms on the samples: 1—PHBV/PCL(1:2), 2—PHBV/PCL(1:2)/CNC(PVP)–5, 3—PHBV/PCL(1:2)/CNC(PVP)–10, 4—PHBV/PCL(1:2)/CNC(PVP)–15 (**a**); 5—PHBV/PCL(1:2)/CNC(PAM)–5, 6—PHBV/PCL(1:2)/CNC(PAM)–10, 7—PHBV/PCL(1:2)/CNC(PAM)–15 (**b**).

**Figure 7 polymers-14-00340-f007:**
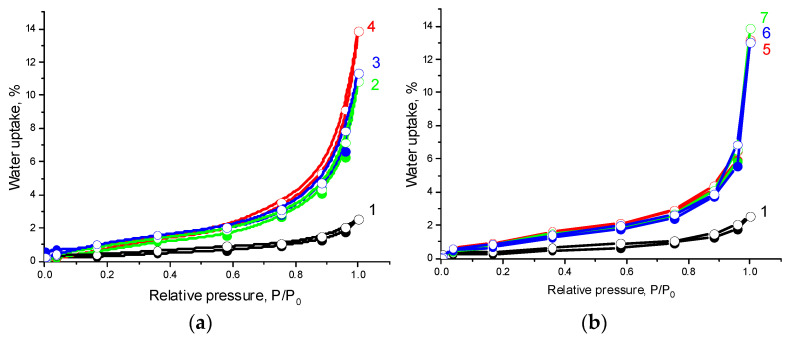
Water vapor adsorption (filled symbols) and desorption (open symbols) isotherms at 25 °C on the samples: 1—PHBV/PCL(1:1), 2—PHBV/PCL(1:1)/CNC(PVP)–5, 3—PHBV/PCL(1:1)/CNC(PVP)–10, 4—PHBV/PCL(1:1)/CNC(PVP)–15 (**a**); 5—PHBV/PCL(1:1)/CNC(PAM)–5, 6—PHBV/PCL(1:1)/CNC(PAM)–10, 7—PHBV/PCL(1:1)/CNC(PAM)–15 (**b**).

**Figure 8 polymers-14-00340-f008:**
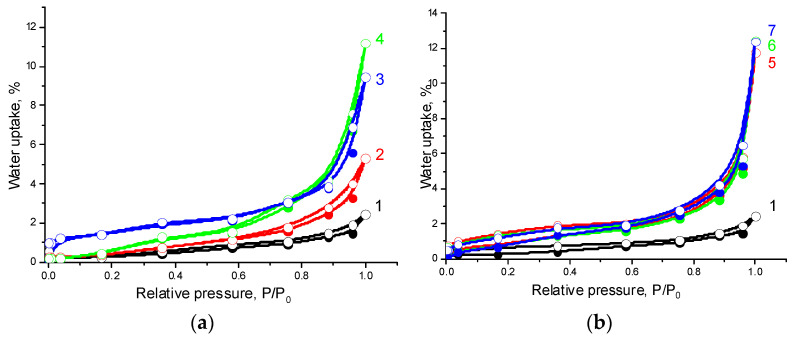
Water vapor adsorption (filled symbols) and desorption (open symbols) isotherms at 25 °C on the samples: 1—PHBV/PCL(1:2), 2—PHBV/PCL(1:2)/CNC(PVP)–5, 3—PHBV/PCL(1:2)/CNC(PVP)–10, 4—PHBV/PCL(1:2)/CNC(PVP)–15 (**a**); 5—PHBV/PCL(1:2)/CNC(PAM)–5, 6—PHBV/PCL(1:2)/CNC(PAM)–10, 7—PHBV/PCL(1:2)/CNC(PAM)–15 (**b**).

**Figure 9 polymers-14-00340-f009:**
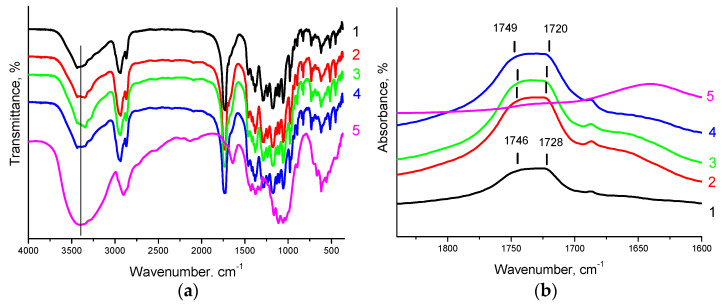
FTIR spectra of the PHBV/PLC(1:1) mixture (1); PHBV/PCL(1:1)/CNC(PVP)–5 (2), PHBV/PCL(1:1)/CNC(PVP)–10 (3), PHBV/PCL(1:1)/CNC(PVP)–15 (4) composites; the neat CNC (5): (**a**) in the 4000–400 cm^−1^ range; (**b**) in the 1850–1600 cm^−1^ range.

**Figure 10 polymers-14-00340-f010:**
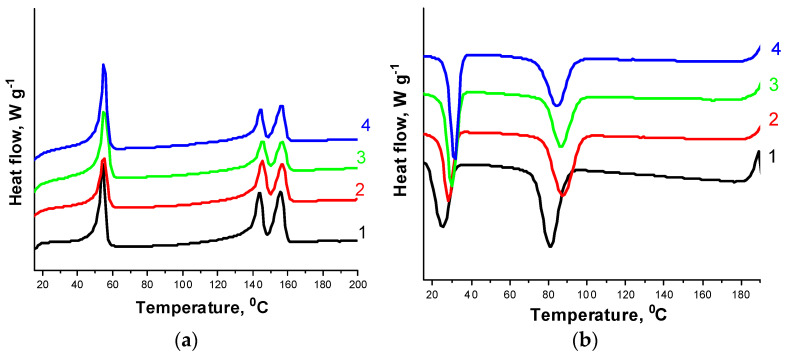
DSC curves for the PHBV/PCL(1:1) mixture (1); PHBV/PCL(1:1)/CNC(PVP)–5 (2), PHBV/PCL(1:1)/CNC(PVP)–10 (3), and PHBV/PCL(1:1)/CNC(PVP)–15 (4) composites: at heating (**a**); at cooling (**b**).

**Figure 11 polymers-14-00340-f011:**
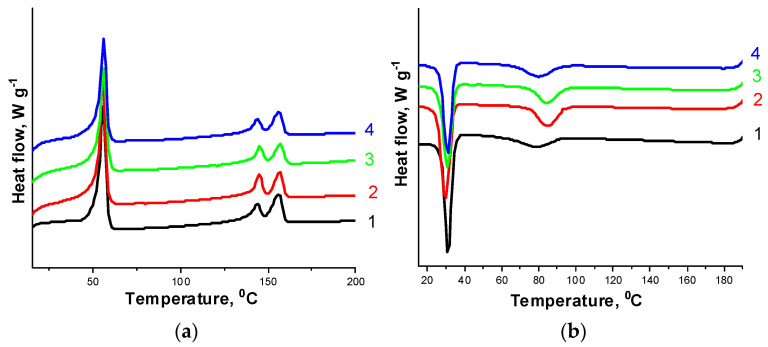
DSC curves for the PHBV/PCL(1:2) mixture (1); PHBV/PCL(1:2)/CNC(PVP)–5 (2), PHBV/PCL(1:2)/CNC(PVP)–10 (3), and PHBV/PCL(1:2)/CNC(PVP)–15 (4) composites: at heating (**a**); at cooling (**b**).

**Figure 12 polymers-14-00340-f012:**
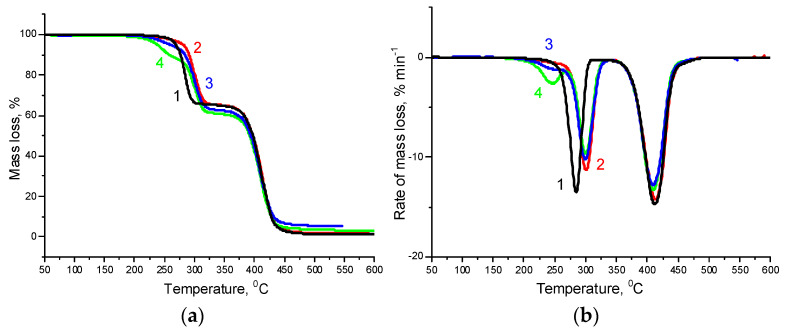
TG (**a**) and DTG (**b**) curves for the PHBV/PCL(1:2) mixture (1); PHBV/PCL(1:2)/CNC(PVP)–5 (2), PHBV/PCL(1:2)/CNC(PVP)–10 (3), and PHBV/PCL(1:2)/CNC(PVP)–15 (4) composites.

**Figure 13 polymers-14-00340-f013:**
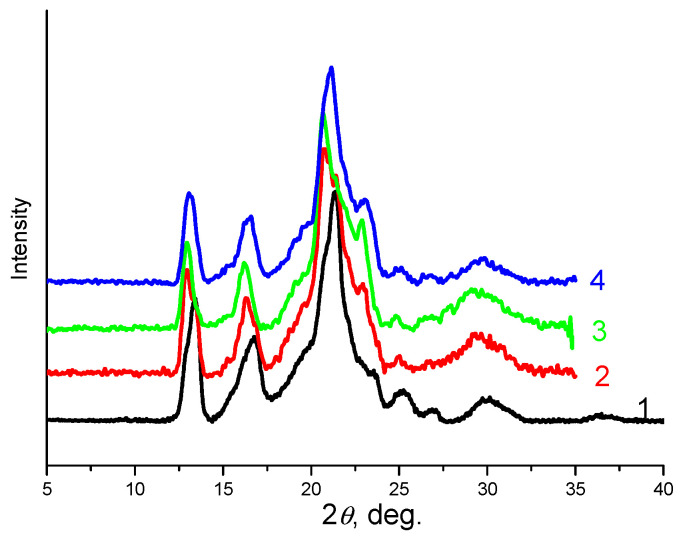
X-ray diffraction patterns of the PHBV/PCL(1:1) mixture (1); PHBV/PCL(1:1)/CNC(PVP)–5 (2), PHBV/PCL(1:1)/CNC(PVP)–10 (3), and PHBV/PCL(1:1)/CNC(PVP)–15 (4) composites.

**Figure 14 polymers-14-00340-f014:**
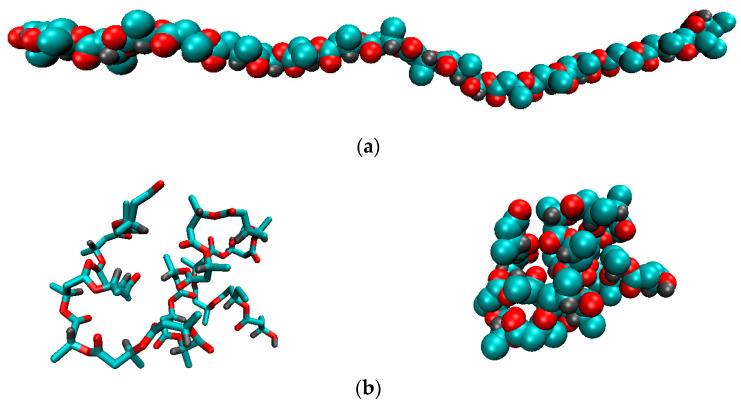
The initial configuration of the PHBV (**a**) (radius of gyration R_g_ = 8.8 nm) and after 1 ns simulation in vacuum (**b**).

**Figure 15 polymers-14-00340-f015:**
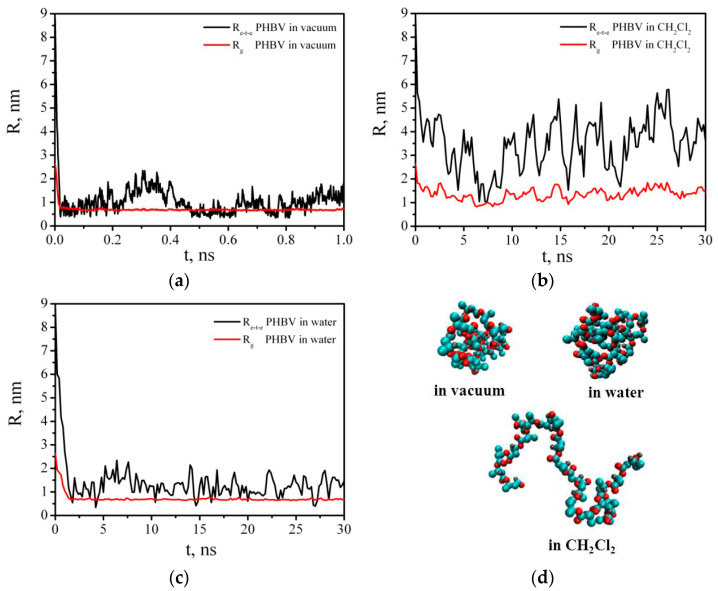
Time dependences of the end-to-end distance (R_e-t-e_) and radius of gyration (R_g_) of the PHBV in vacuum (**a**), DCM (**b**), and water (**c**); instant snapshots of the PHBV chain conformations in vacuum, DCM, and water (**d**).

**Figure 16 polymers-14-00340-f016:**
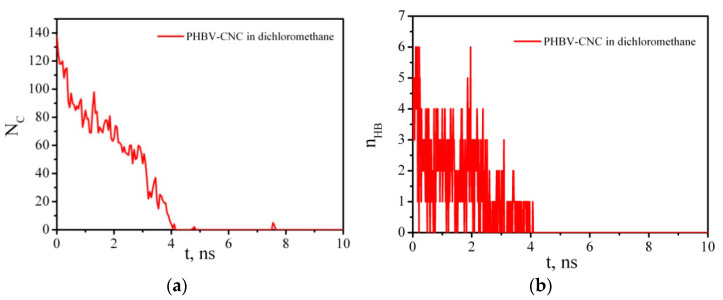
Time dependence of the number of contacts (**a**) and PHBV–CNC hydrogen bonds (**b**); instant snapshots of mutual arrangement of PHBV and CNC in time (**c**) (DCM molecules are not shown for clarity).

**Figure 17 polymers-14-00340-f017:**
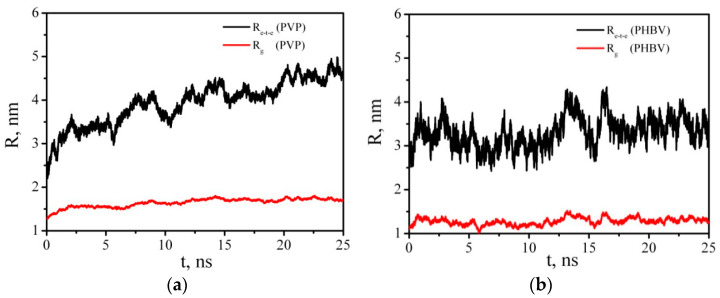
Time dependence of the end-to-end distance R_e-t-e_ and radius of gyration R_g_ of PVP (**a**) and PHBV (**b**).

**Figure 18 polymers-14-00340-f018:**
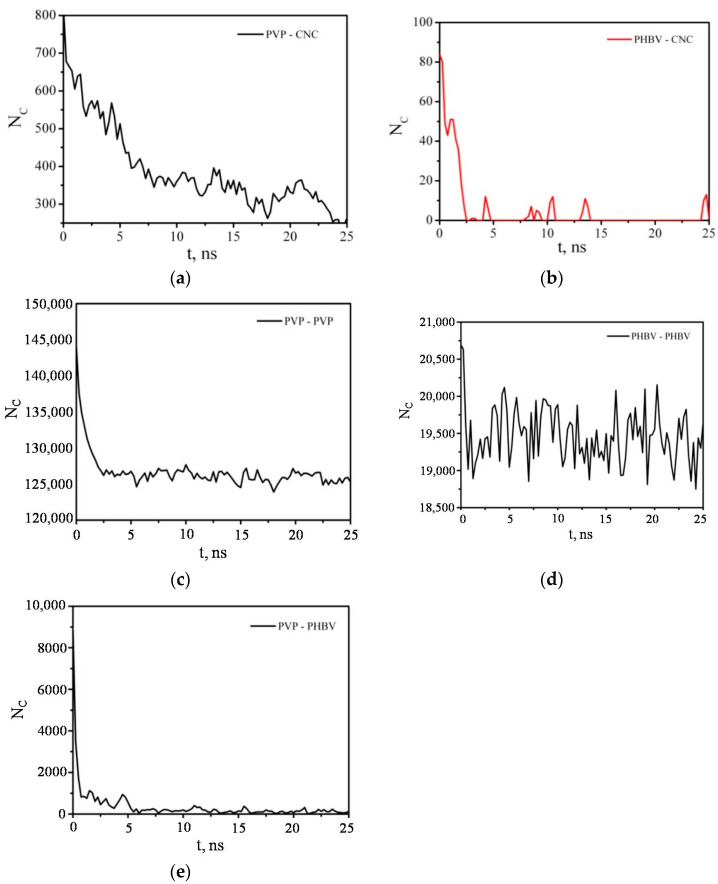
Time dependence of the number of contacts N_c_ between any pair of PVP–CNC (**a**), PHBV–CNC (**b**), PVP–PVP (**c**), PHBV–PHBV (**d**), and PVP–PHBV (**e**) atoms within 0.5 nm.

**Figure 19 polymers-14-00340-f019:**
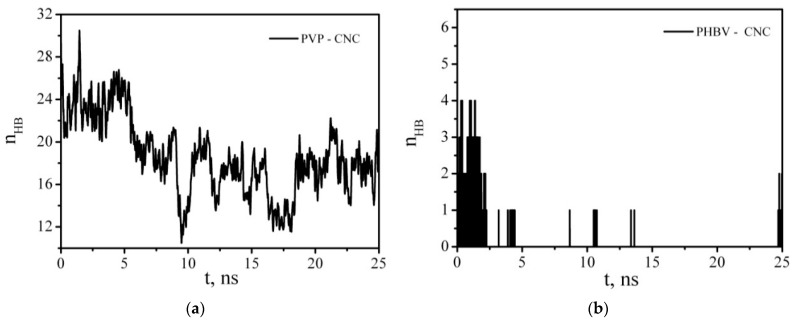
Time dependence of the number of PVP–CNC (**a**) and PHBV–CNC (**b**) hydrogen bonds.

**Figure 20 polymers-14-00340-f020:**
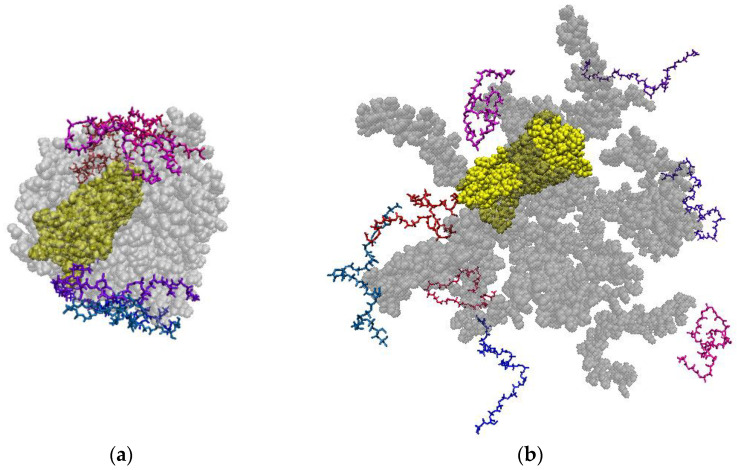
Snapshots of the PHBV/PVP/CNC composite at the beginning (**a**) and at the end (**b**) of the simulation. The PVP macromolecules are shown in grey, PHBV ones in two colors (pink and navy) and the CNC particle is in yellow (DCM molecules are not shown for clarity).

**Figure 21 polymers-14-00340-f021:**
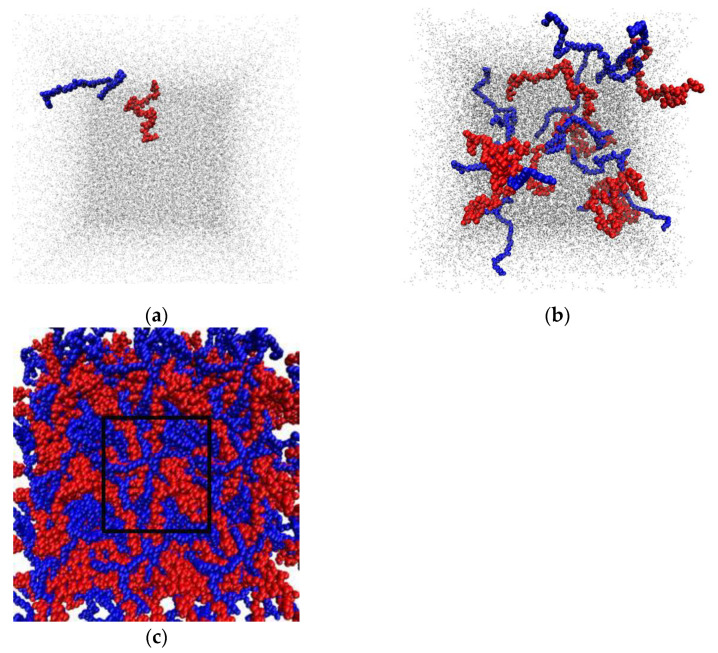
Snapshots of the systems under study at the end of the simulation: System 1 (**a**); System 2 (**b**); System 3 (**c**). PCL is shown in blue, PHBV in red, and DCM in grey.

**Figure 22 polymers-14-00340-f022:**
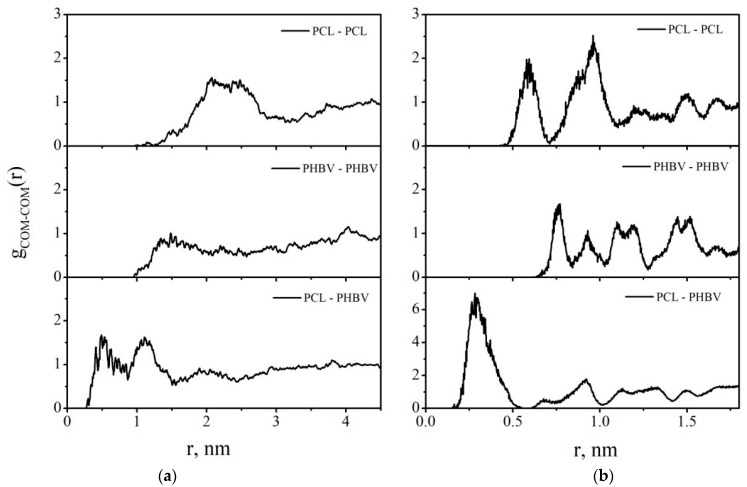
The radial distribution functions between the centers of mass of the polymer macromolecules in System 2 (**a**) and System 3 (**b**).

**Figure 23 polymers-14-00340-f023:**
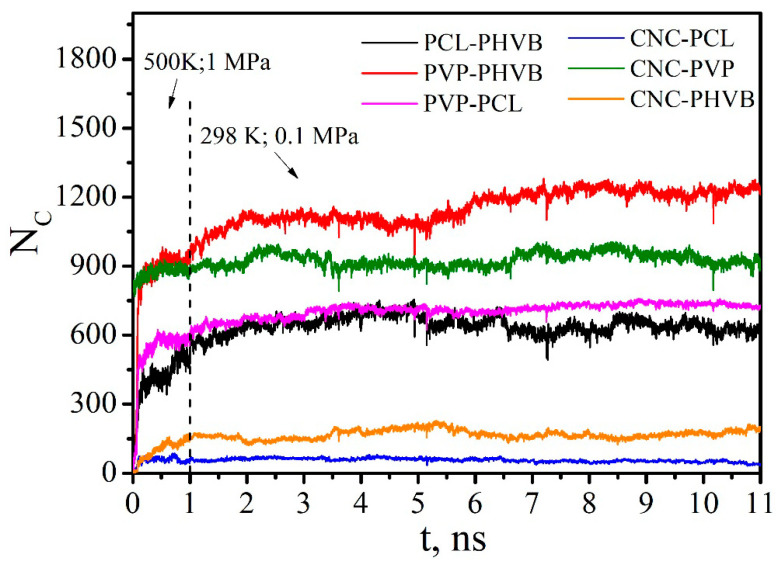
The number of close contacts between atoms of the components within a distance of 0.5 nm.

**Figure 24 polymers-14-00340-f024:**
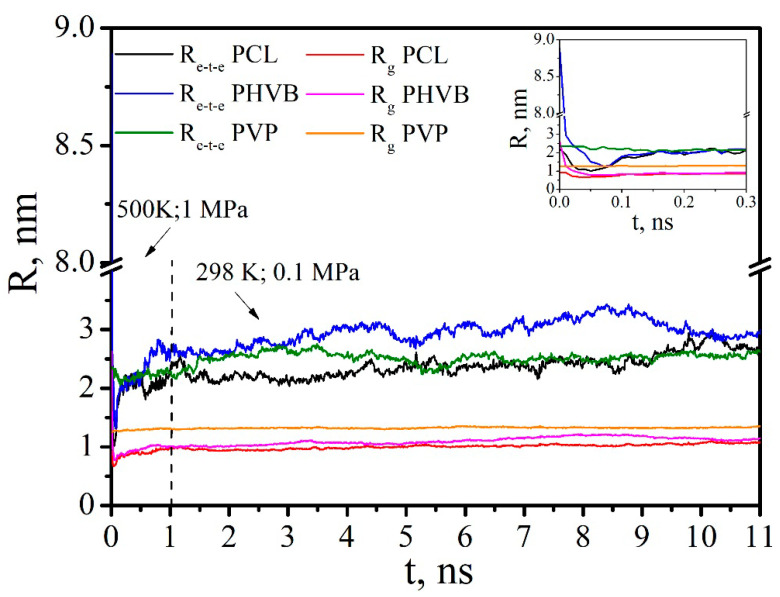
The time dependence of the end-to-end distance and radius of gyration for PCL, PHVB, and PVP.

**Figure 25 polymers-14-00340-f025:**
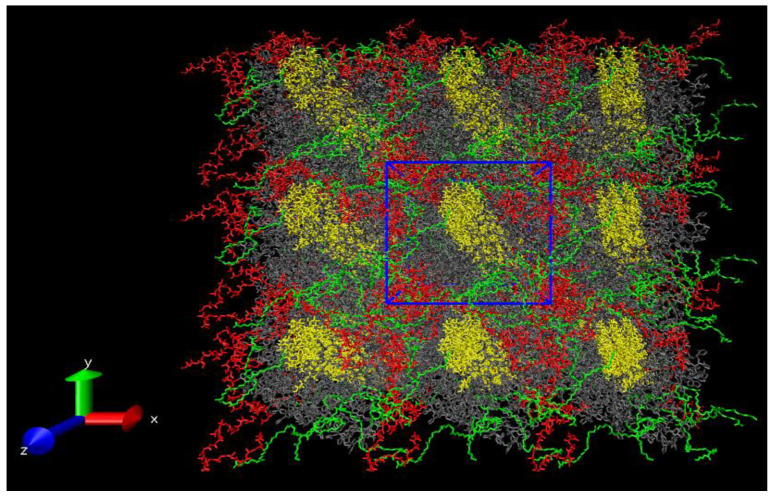
Snapshot of the final configuration of the PHVB/PCL/PVP/CNC composite with periodic images: PVP is colored in grey; CNC is colored in yellow; PHBV is colored in red; PCL is colored in green.

**Figure 26 polymers-14-00340-f026:**
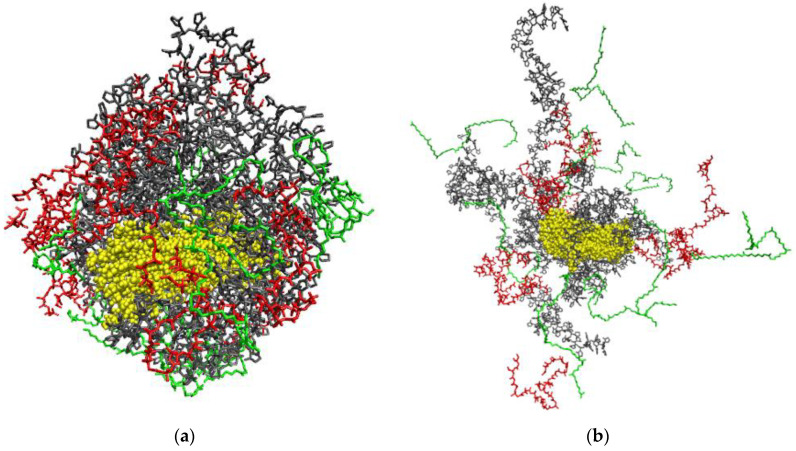
Snapshot of the initial (**a**) and final (**b**) configuration of the PHVB/PCL/PVP/CNC composite in DCM: PVP is colored grey; CNC is colored yellow; PHBV is colored red; PCL is colored green. DCM molecules are not depicted for clarity.

**Figure 27 polymers-14-00340-f027:**
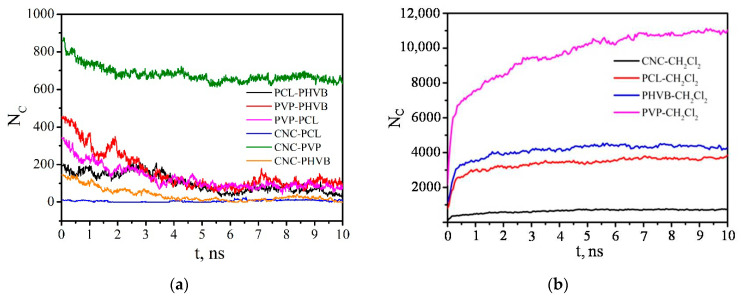
The time dependence of the number of close contacts between any pair of atoms of CNC–polymer, polymer–polymer (**a**), and CNC-DCM, polymer–DCM (**b**).

**Figure 28 polymers-14-00340-f028:**
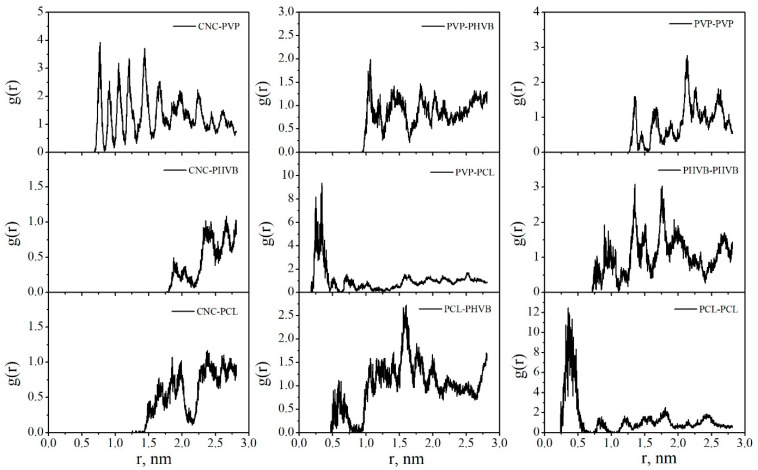
The radial distribution functions between the centers of mass of the composite components in vacuum.

**Figure 29 polymers-14-00340-f029:**
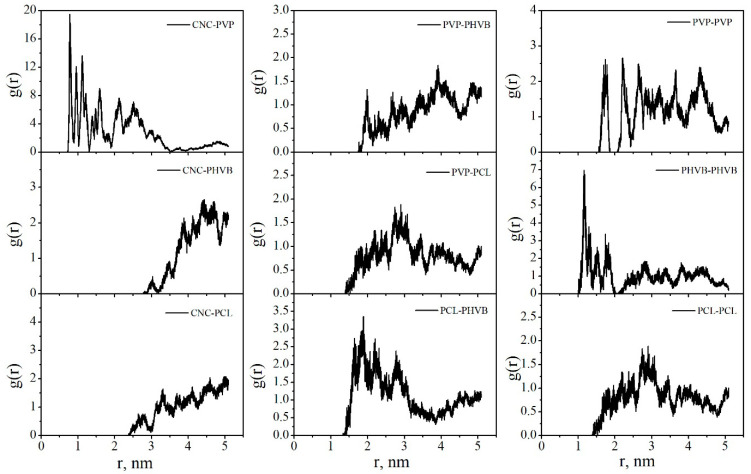
The radial distribution functions between the centers of mass of the composite components in DCM.

**Table 1 polymers-14-00340-t001:** Specific surface area and water vapor sorption on PHBV/PCL and PHBV/PCL/CNC films.

Sample	BET Specific Surface Area, m^2^ g^−1^	Maximum Water Vapor Sorption, %
PHBV/PCL(1:1)	4.9	2.6
PHBV/PCL(1:1)/CNC(PVP)–5	7.8	10.7
PHBV/PCL(1:1)/CNC(PVP)–10	8.5	11.3
PHBV/PCL(1:1)/CNC(PVP)–15	18.7	13.7
PHBV/PCL(1:1)/CNC(PAM)–5	7.2	13.1
PHBV/PCL(1:1)/CNC(PAM)–10	11.4	13.3
PHBV/PCL(1:1)/CNC(PAM)–15	19.0	13.8
PHBV/PCL(1:2)	3.8	2.5
PHBV/PCL(1:2)/CNC(PVP)–5	7.3	5.3
PHBV/PCL(1:2)/CNC(PVP)–10	7.7	9.5
PHBV/PCL(1:2)/CNC(PVP)–15	12.7	11.2
PHBV/PCL(1:2)/CNC(PAM)–5	11.6	11.7
PHBV/PCL(1:2)/CNC(PAM)–10	17.7	12.3
PHBV/PCL(1:2)/CNC(PAM)–15	24.2	12.3

**Table 2 polymers-14-00340-t002:** The temperatures of melting (*T*_m_), crystallization (*T*_cryst_), the heat of fusion (Δ*H*_m_), the degree of crystallinity (*χ*_c_) of PHBV and PCL in mixtures (1:1 and 1:2) and in composites with CNCs.

Sample	*T*_m PCL_, °C	*T*_m1 PHBV_, °C	*T*_m2 PHBV_, °C	*T*_cryst PCL_, °C	*T*_cryst PHBV_, °C	Δ*H*_m PCL_, J g^−1^	Δ*H*_m PHBV_, J g^−1^	*χ*_c PCL_, %	*χ*_c PHBV_, %
PHBV	-	145.7	156.5	-	88.6	-	46.5	-	31.7
PCL	55.6	-	-	24.1	-	76.0	-	48.4	-
PHBV/PCL (1:1)	54.3	144.0	155.8	29.2	84.9	20.9	39.7	26.6	54.1
PHBV/PCL (1:1)/CNC(PVP)–5	54.9	145.3	156.9	28.2	87.6	15.8	28.5	21.2	40.9
PHBV/PCL (1:1)/CNC(PVP)–10	55.3	145.3	156.9	30.0	86.5	21.5	18.9	30.4	28.6
PHBV/PCL (1:1)/CNC(PVP)–15	55.1	144.3	156.3	31.4	84.4	22.3	22.2	33.4	35.6
PHBV/PCL (1:1)/CNC(PAM)–5	55.6	144.5	156.7	31.4	82.8	18.0	18.1	24.1	26.0
PHBV/PCL (1:1)/CNC(PAM)–10	55.1	143.9	156.3	32.0	81.8	20.8	24.6	29.4	37.3
PHBV/PCL (1:1)/CNC(PAM)–15	54.7	143.6	155.8	31.9	84.7	23.2	22.6	34.8	36.3
PHBV/PCL (1:2)	55.7	143.9	156.1	31.1	78.7	30.3	19.3	29.2	39.5
PHBV/PCL (1:2)/CNC(PVP)–5	55.7	144.9	156.8	29.7	84.8	32.2	13.9	32.4	29.9
PHBV/PCL (1:2)/CNC(PVP)–10	56.0	145.2	157.0	30.7	84.5	30.0	11.9	31.8	27.1
PHBV/PCL (1:2)/CNC(PVP)–15	55.8	143.9	156.3	30.9	80.0	29.5	13.2	33.1	31.8
PHBV/PCL (1:2)/CNC(PAM)–5	55.9	143.9	156.1	31.7	79.8	30.8	14.9	31.0	32.1
PHBV/PCL (1:2)/CNC(PAM)–10	55.6	142.9	155.8	31.6	76.4	33.3	13.5	35.4	30.7
PHBV/PCL (1:2)/CNC(PAM)–15	56.3	144.4	156.4	30.7	79.9	27.8	12.9	31.2	31.1

**Table 3 polymers-14-00340-t003:** Thermal stability of the PHBV/PCL mixtures and PHBV/PCL/CNC composites.

Sample	The First Stage of Thermal Decomposition			The Second Stage of Thermal Decomposition			Total Mass Loss, %
	^1^*T*_on_, °C	*T*_max_, °C	Mass Loss, %	^1^*T*_on_, °C	^2^*T*_max_, °C	Mass Loss, %	
PHBV	261.6	275.2	98.6	-	-	-	98.6
PCL	-	-	-	343.7	412.0	96.5	96.5
PHBV/PCL(1:1)	265.0	279.9	60.7	384.8	410.0	38.3	99.0
PHBV/PCL(1:1)/CNC(PVP)–5	278.1	296.4	52.5	386.5	409.7	45.0	97.5
PHBV/PCL(1:1)/CNC(PVP)–10	286.2	298.0	40.7	386.9	407.9	45.3	96.3
PHBV/PCL(1:1)/CNC(PVP)–15	276.2	295.0	50.7	386.7	410.8	44.0	94.7
PHBV/PCL(1:1)/CNC(PAM)–5	278.0	296.6	50.1	387.5	409.0	45.9	96.0
PHBV/PCL(1:1)/CNC(PAM)–10	279.6	297.9	52.9	386.8	408.1	41.6	94.5
PHBV/PCL(1:1)/CNC(PAM)–15	280.0	298.0	53.0	387.0	408.0	42.0	95.0
PHBV/PCL(1:2)	270.0	284.7	34.5	389.5	412.0	64.4	98.9
PHBV/PCL(1:2)/CNC(PVP)–5	283.4	300.8	34.9	389.6	413.3	63.1	98.0
PHBV/PCL(1:2)/CNC(PVP)–10	288.6	299.9	27.2	387.7	410.1	58.1	97.1
PHBV/PCL(1:2)/CNC(PVP)–15	278.4	299.5	37.6	387.1	409.5	57.0	94.6
PHBV/PCL(1:2)/CNC(PAM)–5	280.1	298.7	36.2	387.3	409.5	59.5	95.7
PHBV/PCL(1:2)/CNC(PAM)–10	282.1	299.8	37.2	388.5	411.4	59.6	96.8
PHBV/PCL(1:2)/CNC(PAM)–15	282.0	300.0	37.0	389.0	411.0	60.0	97.0

^1^*T*_on_ is the temperature of the onset of thermal decomposition. ^2^
*T*_max_ is temperature of the maximum rate of thermal decomposition.

**Table 4 polymers-14-00340-t004:** The total crystallinity of PHBV and PCL polymers in the composites with CNCs.

Sample	The Total Crystallinity of PHBV and PCL, %
PHBV/PCL(1:1)	43.8
PHBV/PCL(1:1)/CNC(PVP)–5	35.3
PHBV/PCL(1:1)/CNC(PVP)–10	35.1
PHBV/PCL(1:1)/CNC(PVP)–15	37.6
PHBV/PCL(1:1)/CNC(PAM)–5	33.3
PHBV/PCL(1:1)/CNC(PAM)–10	22.6
PHBV/PCL(1:1)/CNC(PAM)–15	41.4
PHBV/PCL(1:2)	47.2
PHBV/PCL(1:2)/CNC(PVP)–5	49.0
PHBV/PCL(1:2)/CNC(PVP)–10	51.8
PHBV/PCL(1:2)/CNC(PVP)–15	49.1
PHBV/PCL(1:2)/CNC(PAM)–5	39.5
PHBV/PCL(1:2)/CNC(PAM)–10	38.6
PHBV/PCL(1:2)/CNC(PAM)–15	38.9

**Table 5 polymers-14-00340-t005:** Tensile properties of the PHBV/PCL mixtures and PHBV/PCL/CNC composites ^1^.

Sample	Ultimate Tensile Strength (σ_max_), MPa	Tensile Strength at Break (σ_b_), MPa	Elongation at Break (ε), %	Young’s Modulus (*E*), MPa
PHBV/PCL(1:1)	11.7	11.7	8.7	350
PHBV/PCL(1:1)/CNC(PVP)–5	11.7	10.7	82.0	330
PHBV/PCL(1:1)/CNC(PVP)–10	7.1	7.1	2.0	460
PHBV/PCL(1:1)/CNC(PVP)–15	3.2	3.2	3.0	340
PHBV/PCL(1:1)/CNC(PAM)–5	3.7	3.7	4.0	280
PHBV/PCL(1:1)/CNC(PAM)–10	4.2	4.2	5.6	280
PHBV/PCL(1:1)/CNC(PAM)–15	4.3	4.3	7.0	220
PHBV/PCL(1:2)	17.3	17.0	340	260
PHBV/PCL(1:2)/CNC(PVP)–5	11.5	11.5	16.4	250
PHBV/PCL(1:2)/CNC(PVP)–10	10.1	9.4	3.4	480
PHBV/PCL(1:2)/CNC(PVP)–15	7.9	7.9	8.9	360
PHBV/PCL(1:2)/CNC(PAM)–5	9.5	9.5	5.3	430
PHBV/PCL(1:2)/CNC(PAM)–10	10.4	10.4	7.6	510
PHBV/PCL(1:2)/CNC(PAM)–15	10.0	10.0	9.3	360

^1^ The obtained values of the elongation at break and stress are within ±15%, Young’s modulus is within ±10%.

**Table 6 polymers-14-00340-t006:** Simulation details of the systems: number of molecules, macromolecules or particles (N), simulation time (t), and edge length of the simulation boxes (L).

System	N (CNC)	N (PVP)	N (PHBV)	N (DCM)	N (Water)	t, ns	L, nm
1	-	1	-	10,000	-	30	9.75
2	-	-	1	10,000	-	30	9.72
3	-	-	1	-	24,000	30	8.48
4	-	1	1	10,000	-	50	9.73
5	1	1	-	18,000	-	20	11.85
6	1	-	1	-	-	20	11.95
7	1	13	-	-	-	30	20.00
8	1	13	8	10,000	-	20	20.00

**Table 7 polymers-14-00340-t007:** Simulation details of the systems: number of molecules, macromolecules or particles (N), simulation time (t), and edge length of the simulation boxes (L).

System	N (PCL)	N (PHBV)	N (DCM)	t, ns	L, nm
1	1	1	10,000	10	9.73
2	10	10	10,000	10	9.75
3	10	10	-	10	3.58

**Table 8 polymers-14-00340-t008:** The values of the end-to-end distance R_e-t-e_ and radius of gyration R_g_ for PCR and PHBV, averaged over the last 5 ns of the simulation.

System	PCL		PHBV	
	R_e-t-e_, nm	R_g_, nm	R_e-t-e_, nm	R_g_, nm
1	5.33 ± 1.02	1.79 ± 0.23	3.70 ± 0.96	1.42 ± 0.19
2	4.34 ± 0.30	1.57 ± 0.06	3.26 ± 0.37	1.28 ± 0.07
3	3.28 ± 0.06	1.29 ± 0.02	2.85 ± 0.06	1.15 ± 0.01

**Table 9 polymers-14-00340-t009:** The number of close contacts between any pair of the polymer atoms within 0.5 nm averaged over the last 5 ns of the simulation.

System	N_C_ (PCL–PHBV)	N_C_ (PCL–PCL)	N_C_ (PHBV–PHBV)
1	0	656 ± 16	2397 ± 118
2	167 ± 88	6742 ± 88	24,319 ± 451
3	6477 ± 145	10,943 ± 149	33,450 ± 284

**Table 10 polymers-14-00340-t010:** The end-to-end distance R_e-t-e_ and radius of gyration R_g_ of the polymers (as individual macromolecules and as a part of the composite) in vacuum, water, and DCM.

Medium		PCL	PHBV	PVP
Vacuum (individual molecule)	R_e-t-e_, nm	0.63 ± 0.10	1.01 ± 0.43	1.23 ± 0.18
	R_g_, nm	0.69 ± 0.02	0.69 ± 0.04	1.10 ± 0.01
Water (individual molecule)	R_e-t-e_, nm	1.55 ± 0.63	1.24 ± 0.40	1.74 ± 0.20
	R_g_, nm	0.71 ± 0.08	0.68 ± 0.03	1.11 ± 0.01
DCM (individual molecule)	R_e-t-e_, nm	3.78 ± 1.34	3.49 ± 1.05	5.61 ± 1.07
	R_g_, nm	1.46 ± 0.26	1.35 ± 0.25	1.97 ± 0.23
Vacuum (in the composite)	R_e-t-e_, nm	2.59 ± 0.19	3.06 ± 0.14	2.53 ± 0.06
	R_g_, nm	1.06 ± 0.5	1.16 ± 0.03	1.33 ± 0.01
DCM (in the composite)	R_e-t-e_, nm	3.73 ± 0.19	2.99 ± 0.21	4.65 ± 0.19
	R_g_, nm	1.45 ± 0.03	1.20 ± 0.03	1.73 ± 0.02

**Table 11 polymers-14-00340-t011:** Average number of hydrogen bonds formed by cellulose with the polymers in vacuum and in DCM at 298 K and 0.1 MPa.

<nHB>	Vacuum	DCM
CNC–PCL	0.63 ± 0.12	0.12 ± 0.39
CNC–PHVB	2.9 ± 0.3	0.53 ± 0.67
CNC–PVP	38.9 ± 0.5	34.7 ± 3.9

## Data Availability

The data presented in this study are available on request from the corresponding author.

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
