# Peer review of "Properties of Poly(3-hydroxybutyrate-co-3-hydroxyvalerate)/Polycaprolactone Polymer Mixtures Reinforced by Cellulose Nanocrystals: Experimental and Simulation Studies"

_polymers, 2022, doi:10.3390/polym14020340_

Round 1

Reviewer 1 Report

The article entitled "Properties of Poly (3-hydroxybutyrate-co-3-hydroxyvalerate) / Polycaprolactone Polymer Mixtures Reinforced by Cellulose Nanocrystals: Experimental and Simulation Studies" is a summary of a very comprehensive research in the field of modification of biodegradable polymers. In my opinion, the authors have collected a very extensive research material, which, after a few corrections, can be published in the form of a scientific article.

There are no objections to the scientific value of the presented work, the presentation of the results as well as the formulated conclusions are at a high level. The only reservation concerns a very broad form of presentation of the results, which requires the reader to analyze the text very thoroughly. Some minor suggestions are presented below:

  1. The number of SEM photos is too large, while the comparisons themselves are very valuable, however, only the most characteristic ones should be included in the main article text, while the rest of the photos can be removed in an additional Suplementary data file.
  2. The number of POM pictures should also be reduced
  3. DSC charts markind is unclear, with such a number of curves, the single curve should have a separate caption, while some of the charts should also be omitted, because they do not provide new information, since the enthalpy or crystallinity values are included in the table. The same remarks refer to TGA plots.
  4. Tensile stress-strain should be replace to additional file

Author Response

Dear Editors and Reviewers,

I appreciate your taking the time to read our article. Thank you very much for the opportunity to revise and resubmit our paper taking into account the reviews received. Here are responses to the reviewer's comments.

Reviewer 1

  1. The number of SEM photos is too large, while the comparisons themselves are very valuable, however, only the most characteristic ones should be included in the main article text, while the rest of the photos can be removed in an additional Suplementary data file.
  2. The number of POM pictures should also be reduced
  3. DSC charts markind is unclear, with such a number of curves, the single curve should have a separate caption, while some of the charts should also be omitted, because they do not provide new information, since the enthalpy or crystallinity values are included in the table. The same remarks refer to TGA plots.
  4. Tensile stress-strain should be replace to additional file

Author response:

In the main body of the revised manuscript, the only most important results have been kept, the other figures have been moved to Supplementary Material.

Reviewer 2 Report

This paper presented an investigation of PHBV/PCL and CNC reinforced composites. The authors first reviewed the state of the art of PHBV and PCL polymers. Then, the materials, process, and testing methods were presented with details. The authors reported more than an adequate amount of results with some scientific discussions. The reviewer suggests revising this paper before publishing it. 

  1. The paper has too many figures as results in the current draft. There are 50 figures. Could the authors split the paper into two papers or move most of the figures to the supporting materials section? Please only keep the most important results in the revised draft.
  2. Some of the figures have different format than the others. For example, figures 8 and 9 have black boxes around the figures. Please keep them in the same format.

Author Response

I appreciate your taking the time to read our article. Thank you very much for the opportunity to revise and resubmit our paper taking into account the reviews received. Here are responses to the reviewer's comments.

Reviewer 2

  1. The paper has too many figures as results in the current draft. There are 50 figures. Could the authors split the paper into two papers or move most of the figures to the supporting materials section? Please only keep the most important results in the revised draft.
  2. Some of the figures have different format than the others. For example, figures 8 and 9 have black boxes around the figures. Please keep them in the same format.

Author response:

In the main body of the revised manuscript, the only most important results have been kept, the other figures have been moved to Supplementary Material. All figures have been kept in the same format.